# The Circulating Level of Klotho Is Not Dependent upon Physical Fitness and Age-Associated Methylation Increases at the Promoter Region of the Klotho Gene

**DOI:** 10.3390/genes14020525

**Published:** 2023-02-19

**Authors:** Dora Aczel, Ferenc Torma, Matyas Jokai, Kristen McGreevy, Anita Boros, Yasuhiro Seki, Istvan Boldogh, Steve Horvath, Zsolt Radak

**Affiliations:** 1Research Institute of Sport Science, Hungarian University of Sport Science, 1123 Budapest, Hungary; 2Sports Neuroscience Division, Advanced Research Initiative for Human High Performance (ARIHHP), Faculty of Health and Sport Sciences, University of Tsukuba, Tsukuba 305-8574, Japan; 3Department of Biostatistics, Fielding School of Public Health, University of California Los Angeles, Los Angeles, CA 90095, USA; 4Faculty of Sport Sciences, Waseda University, Tokorozawa 2-579-15, Japan; 5Department of Microbiology and Immunology, University of Texas Medical Branch at Galveston, Galveston, TX 77555, USA

**Keywords:** aging, klotho, methylation, epigenetic clock, physical fitness

## Abstract

(1) Background: Higher levels of physical fitness are believed to increase the physiological quality of life and impact the aging process with a wide range of adaptive mechanisms, including the regulation of the expression of the age-associated klotho (KL) gene and protein levels. (2) Methods: Here, we tested the relationship between the DNA methylation-based epigenetic biomarkers PhenoAge and GrimAge and methylation of the promoter region of the KL gene, the circulating level of KL, and the stage of physical fitness and grip force in two groups of volunteer subjects, trained (TRND) and sedentary (SED), aged between 37 and 85 years old. (3) Results: The circulating KL level is negatively associated with chronological age in the TRND group (r = −0.19; *p* = 0.0295) but not in the SED group (r = −0.065; *p* = 0.5925). The age-associated decrease in circulating KL is partly due to the increased methylation of the KL gene. In addition, higher plasma KL is significantly related to epigenetic age-deceleration in the TRND group, assessed by the biomarker of PhenoAge (r = −0.21; *p* = 0.0192). (4) Conclusions: The level of physical fitness, on the other hand, does not relate to circulating KL levels, nor to the rate of the methylation of the promoter region of the KL gene, only in males.

## 1. Introduction

Human aging is common and may be considered an irreversible process. However, new research suggests that aging can be slowed or even reversed, at least by eliminating changes in gene activity [1,2]. Klotho (KL), named after the Greek goddess of destiny who was the spinner of the thread of life, is involved in the aging process and may act as an anti-aging hormone in mammals. The KL gene is located in chromosome 13 in humans. It encodes a transmembrane protein with an extracellular domain, a transmembrane domain, and an intracellular domain. KL was first described as a gene that shortens lifespan in both mice and Caenorhabditis elegans [3,4]. The absence of KL in mice does not result in visible phenotypic changes until 3–4 weeks of age but then contributes to aging-like phenotypes and dies around 2 months of age [3,5]. In experimental animals, the lack/defect in the KL gene leads to the development of multiple disorders such as stunted growth, hypogonadotropic hypogonadism, skin atrophy [3], rapid thymic involution [6], osteopenia [7], pulmonary emphysema [8], cognitive impairment [9], and hearing problems [10] similar to human aging and causes a shortened lifespan. Indeed, the lifespan of transgenic mice overexpressing Klotho is prolonged [11]. In humans, three protein isoforms of the KL gene exhibit transmembrane glycoprotein with a molecular weight of 130 kDa, the secreted form with a molecular weight of 70 kDa, and the intracellular form with a molecular weight of around 130 kDa. The circulating KL is the secreted isoform, the level of which tends to decrease after the age of forty. Semba et al. reported a positive association between grip strength and circulating KL levels after adjusting the results for age, sex, education, smoking, physical activity, and chronic diseases [12]. Interestingly enough, the level of circulating KL appears to be complex. In community-dwelling older adults, when the circulating KL was higher than 738 pg/mL, a positive relationship was found between lower walking disabilities. Otherwise, when the KL levels were less than 535 pg/mL, the KL concentration was associated with walking disability, showing the complexity of the relationship of KL with physiological functions [13]. When the interrelationship between circulating KL levels and frailty was studied, it turned out that higher plasma KL concentrations were associated with a lower probability of frailty [14]. Overall, the data of a systemic review suggest that physical activity and exercise elevate the level of circulating KL, which is in a negative correlation with the incidence of lifestyle-related diseases [15]; meanwhile, as age progresses, KL protein levels decrease, so it is hypothesized that keeping levels constant can promote healthier aging and modify the incidence of disease [16].

There are different forms of the KL protein, the transmembrane, and the secreted KL, which all perform distinct functions [3,17,18]. Transmembrane KL makes a complex with fibroblast growth factor (FGF) receptors; for example, KL is also an obligate co-receptor for fibroblast growth factor 23 (FGF23). Transmembrane KL has a phosphaturic effect and also regulates the formation of active vitamin D, which is responsible for adequate calcium homeostasis [19,20,21]. In KL- or FGF23-deficient mice, not just phosphate retention but also premature aging syndrome has been shown, highlighting a possible association between phosphate metabolism and aging [3]. Membrane-anchored proteases ADAM10 and ADAM17 can cleave and secrete the extracellular domain of KL into blood/serum, urine, and cerebrospinal fluid [22]. Secreted KL regulates oxidative stress, growth factor receptors, and ion channels [20].

Soluble KL (sKL) functions as a circulating hormone and not only regulates multiple receptors and numerous ion channels, but is inversely associated with mortality [17]. KL also inhibits the insulin/insulin-like growth factor 1 (IGF1) pathway, an evolutionarily conserved method for prolonging life [23]. sKL protein is considered a powerful biomarker of longevity [3] and contributes to anti-aging through calcium and phosphate metabolism regulation [3,24,25], inflammatory process reduction [26], and oxidative stress protection [27]. KL’s tumor suppressive activity was first described in breast cancer [23], then in other solid tumors such as pancreatic cancer, cervical cancer, and melanoma [28]. Although the exact molecular mechanism of KL in suppressing aging-related phenotypes is not well known, it appears that aging results in the methylation-dependent suppression of the KL gene expression, at least in monkeys [29]. Indeed, methylation of DNA has been used to create epigenetic biomarkers of aging such as PhenoAge and GrimAge, which are sensitive markers of mortality and can also show the progress of aging [30,31]. Acute and regular exercise can alter the level of circulating KL [32,33]. However, the relationship between circulating KL and DNA methylation-based epigenetic clocks are unknown. We hypothesize that circulating KL is correlated with the age-associated methylation of the promoter region of the KL gene, and this is one of the reasons for the age-related decline in circulating KL. In addition, exercise status may affect circulatory KL levels.

## 2. Materials and Methods

### 2.1. Subjects

The study was approved by the National Public Health Center in Hungary (25167-6/2019/EÜIG) by the Helsinki Declaration. The subjects were recruited as volunteers who signed a written consent form to participate in the study. In total, 202 subjects between the ages of 37 and 85 were included in the study. A great percentage of the volunteers participated in the World Rowing Masters Regatta in Velence, Hungary. They were considered to be the trained group (TRND): *n* = 131; 80 males: age 59.14 ± 10.8; 51 females: age 57.24 ± 9.4. The results were compared to the sedentary group (SED): *n* = 71; 27 males: age 55.63 ± 13.4; 44 females: age 61.91 ± 10.5. A questionnaire asked subjects about their health issues, educational status, and lifestyle, including training habits (Table 1).

### 2.2. Tests and Blood Collection

Our blood sample collection was performed after 12 h of fasting, 24–48 h after the last exercise session, and before a series of multi-step tests. First, whole blood samples were collected from the cubital vein to EDTA vacutainer tubes for the determination of DNA methylation. The whole blood samples were stored at −70 degrees Celsius. For Klotho protein measurement, blood was taken in vacutainer tubes containing ACDAs. After centrifugation (3000 g, 10 min, 4 degrees Celsius), the serums were separated and tested for blood chemistry, and the remaining serum was stored at −70 degrees Celsius.

### 2.3. Physiology Tests

Relative maximal oxygen uptake (VO2max) is one of the top viability markers. For example, it is well known that a higher level of VO2max is associated with decreased risks of many diseases [34,35]. In our study, we used the Chester step test to estimate VO2max [36]. The Chester step test requires participants to step up and down from a 15-cm-high stool for up to 5 × 2 min with their hands on their hips. After the first 2-min interval and a short pause (5 s), the pace of the steps accelerates. Subjects perform the exercise until their heart rate reaches 80% of their individually estimated maximum heart rate, according to the Tanaka protocol (208 − 0.7 × age), or until the 10-min test is completed. By fitting a straight line to the heart rate values achieved during each interval, the maximum relative aerobic capacity of individuals can be estimated. The mean VO2max value of TRND males was 42.27 ± 9.2 mL/kg/min, while for the females, it was 41.46 ± 8.4 mL/kg/min. The SED male VO2max value was 38.25 ± 7.9 mL/kg/min, and the SED female value was 32.55 ± 6.6 mL/kg/min.

During the maximum hand-grip force measurements, subjects alternately grip the device (CAMRY EH101 dynamometer) with their right and left hand three times repeatedly for approximately 2 s with the maximum force possible.

### 2.4. Klotho ELISA Kit

Plasma levels of KL were measured using the Human Klotho ELISA kit, according to the manufacturer’s instructions (R&D Systems, DuoSet ELISA, Cat #DY5334-05, Minneapolis, MN, USA). The measurements were completed on a 96-well microplate in singlicate (pilot studies have shown no significant differences between measuring duplicates or singlicates). Coefficients of variation were less than 10% for the assay. The range of the assay standard curve was from 7 to 7000 pg/mL. Optical densitometry was interpreted with an ELISA plate reader (Thermo Labsystems Multiskan EX) at 450 nm and 595 nm, and after background subtraction, KL was calculated in pg/mL. The natural logarithm of the resulting values was (ln sKL) used in the statistical analysis.

### 2.5. DNA Methylation and Promoter Region Methylation of KL

Epigenome-wide (more than 850 K CpG sites included) DNA methylation was determined with the Infinium Methylation EPIC array (Illumina Inc., San Diego, CA, USA). 

First, bisulfite conversion was made using 500 ng of genomic DNA, the EZ-96 DNA Methylation MagPrep Kit (Zymo Research, Irvine, CA, USA), and the KingFisher Flex robot (Thermo Fisher Scientific, Breda, the Netherlands). The samples were taken in random order. Bisulfite conversion was completed with the following alterations. Fifteen µL of MagBinding Beads was used for DNA binding. The conversion reagent was held in 16 cycles at 95 °C for 30 s, followed by 1 h incubation at 50 °C. After each cycle, the samples were stored at 4 °C for 10 min. After, bisulfite-treated samples were hybridized to EPIC array (Illumina Inc., San Diego, CA) with the modification of 8 µL of DNA used as starting material. To examine the quality of the DNA methylation data, Meffil and Ewastools packages with R version 4.0.0 were used [37]. Samples were excluded with poor quality criteria set by Illumina or with a call rate < 96% or at least 4% of undetected probes. Probes with a detection *p*-value > 0.01 in at least 10% of the samples were set as undetected. Probes with a bead number < 3 in at least 10% of the samples were also excluded. To determine the methylation level, the minfi R package was used, applying “Noob” normalization [38].

The promoter region’s methylation of KL was executed in the following way. The promoter and promoter flank have 17 CpGs, and the core promoter has 16 CpGs. The 13 CpGs analyzed span the region 33,015,146–33,016,356 on chromosome 13 using Hg38. A total of 36 CpGs were within the KL’s gene and regulatory elements (between 33,006,202 and 33,076,000). The EPIC hg38 manifest was used to determine the genomic locations of measured CpG sites. Here, we monitored and examined 13 CpGs of KL’s promoter region, which were included in the 850 K analysis.

### 2.6. Determination of Epigenetic Clocks

Numerous analyses are available for creating lifespan predictors from DNAm data. They identify groups of individual methylation sites and methylation status, measuring chronological age. One of the first pan-tissue epigenetic clocks was made by Horvath (2013) [39] using 353 CpGs. After, Hannum et al. found 71 CpGs in leukocytes which were highly predictive for age [40]. These estimators predict lifespan after adjusting for chronological age and other risk factors. Here, we use two significant epigenetic clocks from the ‘new era’, PhenoAge, and GrimAge. They were calculated as described earlier [30,31]. Horvath’s online age calculator was used to process the methylation data and calculate the rate/rate of aging (https://dnamage.genetics.ucla.edu/ accessed on 25 May 2020). Briefly, the development and validation of the epigenetic biomarker of phenotypic age, also called DNAm PhenoAge, have been detailed elsewhere by Levine et al. [31]. The phenotypic age was assessed by the National Health and Nutrition Examination Survey III as training data, in which a proportional hazards penalized regression model was used to narrow 42 biomarkers to 9 biomarkers and chronological age; this was subsequently validated in the National Health and Nutrition Examination Survey IV. Next, DNAm PhenoAge was developed by regressing phenotypic age on blood DNA methylation data using the Invecchiare in Chianti (Aging in the Chianti Area) data, which produced an estimate of DNAm PhenoAge based on a weighted sum of 513 CpGs. This measure was subsequently validated using multiple cohorts, aging-related outcomes, and tissues/cells. Finally, the underlying biology of the 513 CpGs and the DNAm PhenoAge measure was examined using a variety of complementary data and various genome annotation tools, including chromatin state analysis and gene ontology enrichment. In general, DNAm PhenoAge can be contrasted against chronological age to infer accelerated/decelerated aging. Individuals whose DNAm PhenoAge exceeds their chronological age are thought to be aging at an accelerated rate, which is in line with findings that these individuals tend to have a higher mortality and morbidity risk. DNAm GrimAge was described by Lu et al. This epigenetic clock predicts an individual’s risk of morbidity and mortality by analyzing methyl groups on the DNA which change with age. DNAm GrimAge is different from DNAm PhenoAge because the GrimAge epigenetic clock is a combination of chronological age, sex, DNAm-based estimator of smoking pack-years (because smoking is a significant risk factor of mortality and morbidity), and DNAm-based surrogate markers of 7 plasma protein variables such as adrenomedullin (ADM), beta-2 microglobulin (B2M), cystatin C (Cystatin C), growth differentiation factor 15 (GDF-15), leptin (Leptin), plasminogen activation inhibitor 1 (PAI-1), and tissue inhibitor metalloproteinases 1 (TIMP-1). These DNAm-based biomarkers (200 CpGs each, total 1030 CpGs) were combined into an epigenetic clock. After, Lu et al. performed a meta-analysis generated from blood samples of 6935 individuals, showing that a biomarker is a more punctual predictor of time-to-death, time-to-coronary heart disease, and time-to-cancer. This epigenetic clock also shows a strong relationship with computed tomography data for fatty liver/excess visceral fat and age-at-menopause [30].

The acceleration of PhenoAge and GrimAge (AgeAccelPheno and AgeAccelGrim) refers to the ‘raw residual’ that results from the deviation between the observed and expected value when comparing the age estimate based on methylation to chronological age. If we talk about acceleration, the value has a negative sign. If we talk about deceleration, the value has a positive sign.

### 2.7. Statistical Analysis

For statistical analysis, Statistica 13 software (TIBCO) was used. Before testing the associations between variables, normality was tested by the Shapiro–Wilks test. If normal distribution was confirmed, Pearson’s correlation coefficient was calculated with KL values included as a dependent variable. The significance level was determined as *p* < 0.05.

Analyzing differential methylation was performed using the DMRcate package in R on methylation beta values. We specified the smoothing lambda to be 1000 with a minimum of 2 CpGs in a region. The False Discovery Rate *p*-value threshold was set to 0.05.

## 3. Results

It was found that the age-associated decrease in KL levels was linked to greater methylation of the promoter region of the KL gene. Our analysis evaluated whether KL gene methylation was significantly altered by exercise status in males after controlling for age.

We confirmed in the earlier results that the circulating level of KL decreases with aging only in the TRND group (r = −0.19; *p* = 0.0295, Figure 1A), but not in the SED (r = −0.065; *p* = 0.5925, Figure 1B).

When we tested Gripmax, we found that in the TRND group, KL was associated with higher grip force (r = 0.24; *p* = 0.0058, Figure 2A), but this correlation ‘disappeared’ in the SED group (r = 0.19; *p* = 0.1142, Figure 2B).

However, when the relationship between the level of physical fitness (VO2max) was tested, it turned out that the higher level of estimated VO2max was not associated either with a higher level of circulating KL protein in the TRND group (r = 0.013; *p* = 0.8807, Figure 3A), or in the SED group (r = 0.048; *p* = 0.7008, Figure 3B).

When the relationship between the circulating KL level and the PhenoAge and GrimAge was examined, data revealed that KL is associated with PhenoAge acceleration in the TRND group only. A higher level of blood KL related to the slowing down of the epigenetic aging assessed by PhenoAge (r = −0.21; *p* = 0.0192, Figure 4A; SED: r = −0.17; *p* = 0.1587, Figure 4B). However, GrimAge acceleration did not show a significant relationship with KL levels in either group (TRND: r = 0.02; *p* = 0.8133, Figure 4C; SED: r = 0.02; *p* = 0.8580, Figure 4D).

## 4. Discussion

It is well established that KL levels decrease with age [26,41]. However, to our knowledge so far, there are no reports on the relationship of KL with the DNA methylation-based epigenetic biomarkers of aging. PhenoAge and GrimAge were developed to assess the lifestyle effects on the progress of aging, morbidity, and mortality [42]. The increasing number of investigations on DNA methylation-based biomarkers are more precise and associated with cardiovascular disease, diabetes, cancer, and mortality [43]. Here, we report that reduced levels of circulating KL are related to PhenoAge acceleration; therefore, it appears that higher KL can decelerate the DNA methylation-based aging process assessed by PhenoAge. It has been reported that PhenoAge is strongly related to age-associated diseases and can be assessed from whole blood, saliva, and organ samples. It is moderately heritable and sensitive to inflammation, interferon, DNA damage, and repair, as well as to transcriptional and translational signaling [31]. In addition, Seki and co-workers recently showed that PhenoAge acceleration is associated with the length of telomeres [44]. The physical fitness-related DNA methylation-based age-decelerating effects are novel but correspond properly with the results of epidemiological studies [45,46,47]. The age-related decline in circulating KL levels is known, and our study has shed light on the fact that this decline could be due to increased methylation of the promoter region in the KL gene. Epigenetic modification of DNA is lifestyle-related, which means that the age-associated increased methylation of KL and the related decline in circulating KL can be attenuated. Our finding is corroborated by a recent paper, which reports that the methylation of KL promoter region was associated with a lower level of circulating KL and enhanced levels of pro-inflammatory markers such as interleukin-10, tumor necrosis factor, and nuclear factor kappa B levels in the blood [48].

The relationship between the poor results of the hand grip test and low KL level was observed earlier [12], and our present data are on the original observation. The analysis of more than 200,000 subjects with different age groups from Germany revealed that the grip strength increased in the third and fourth decade of life and declined after the fourth decade. The presence of cut-off points for low grip strength were 29 kg for men and 18 kg for women [49]. Interestingly, it was found that the shape of the negative association between grip strength and mortality was nearly linear without an impact of cut-off points, underlying the importance of grip strength measurements in large epidemiological studies. In this study, a low KL level was also associated with a poor level of knee extensor strength [50]. Interestingly enough, the circulating KL level does not correlate well with the KL level measured in the quadriceps muscle of humans but relates to quadriceps strength [51]. Therefore, KL regulation appears to be complex, but the circulation KL association with strength values is well established. Because strength could be linked to muscle mass and the level of anabolic hormones, it cannot be excluded that higher levels of these hormones are associated with to KL levels. Indeed, it has been shown that the administration of growth hormone increases the level of circulatory KL [52]. When children with growth hormone deficiency were treated with growth hormone, the increase in growth speed was also significantly linked to circulating KL levels and IGF-1 [53,54]. This discovery raises the possibility that circulating KL is not just released from kidneys, as thought earlier, but rather from pituitary cells, since subjects with growth hormone-producing adenomas show elevated levels of KL, which are robustly decreased following a pituitary operation [54]. Lower circulating KL is associated with numerous human diseases. For example, it increases the possibility of hypertension [55], chronic kidney disease [56], interstitial lung abnormalities [57], increased risk for progression of diabetic retinopathy [58], and Alzheimer’s disease [59]. Shmulevich and co-workers [60] suggest that KL is a tumor-suppressor in breast cancer. The mechanism is based on altering the metabolic profile of cancer cells by the inhibition in mitochondrial activity and the reduction in breast cancer-specific calcium influx. This is probably a result of the regulation of KL in calcium channels, such as transient receptor potential vanilloid 5,6. The metabolic background of breast cancer cell suppression by KL could be due to decreased Ca++-associated decline in ATP levels leading to the activation of AMPK via serine/threonine kinase 11 [60]. Interestingly enough, the systemic effects of regular exercise beneficially affect all of the aforementioned diseases [61,62,63,64,65], suggesting that some of the systemic effects of exercise might be mediated by exercise-induced circulating KL levels. Similarly to the present study, when the circulating KL levels were evaluated in young and master athletes (sprinters and endurance athletes), the data revealed that master sprinters presented better kidney function in relation to endurance athletes and middle-aged untrained peers. Moreover, master athletes showed better antioxidant profiles, lower levels of inflammation and longer telomeres, and greater concentrations of circulating KL than aged-matched controls [66]. It was suggested that KL and sestrin-2 may play a role in exercise-induced kidney function protection [67]. The present study revealed that the level of circulating KL is associated with the level of exercise status and general strength level of the body and is greatly dependent upon exercise-induced methylation of DNA since decreased DNA methylation-based age acceleration assessed by PhenoAge was related to higher levels of circulating KL.

## Figures and Tables

**Figure 1 genes-14-00525-f001:**
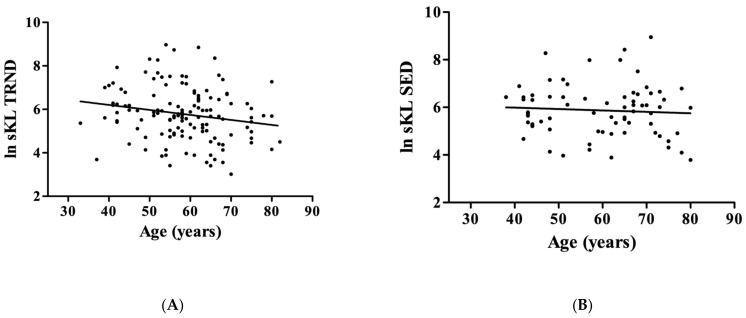
The association between Klotho (KL) and chronological age in the trained (TRND) and in the sedentary (SED) groups. The circulating level of KL showed a negative correlation with chronological age in the TRND group (Panel (**A**); N = 131), but not in the SED group (Panel (**B**); N = 71).

**Figure 2 genes-14-00525-f002:**
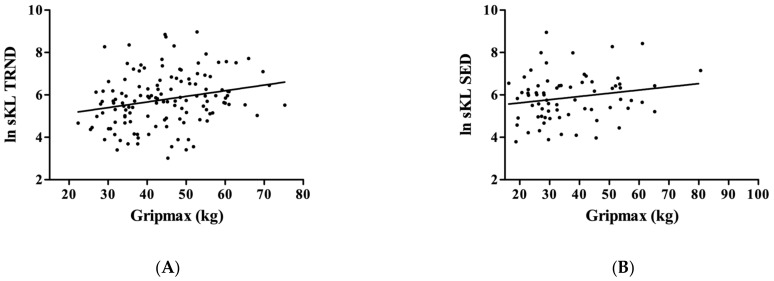
The relationship between Gripmax and KL. The maximal hand grip force showed a significant relationship in the TRND group (Panel (**A**), N = 131), but not in the SED group (Panel (**B**), N = 71).

**Figure 3 genes-14-00525-f003:**
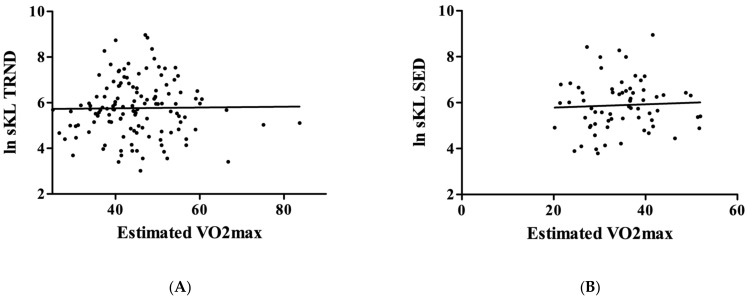
The relationship between estimated VO2max and KL. The maximal oxygen uptake was estimated, and no significant relationship was found in the TRND group (Panel (**A**), N = 131) and the SED group (Panel (**B**), N = 71).

**Figure 4 genes-14-00525-f004:**
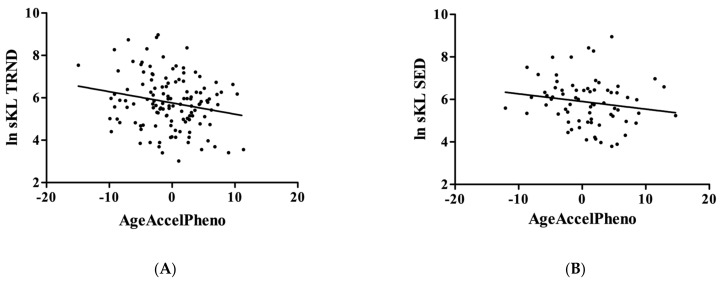
The correlation between KL and DNAmPhenoAge and DNAmGrimAge. The epigenetic aging was calculated on the DNA methylation pattern based on the description of DNAmPhenoAge and DNAmGrimAge. KL is associated with PhenoAge acceleration in the TRND group (Panel (**A**), N = 131), not in the SED group (Panel (**B**), N = 71). However, GrimAge acceleration did not show a significant relationship with KL in either group (Panel (**C**), N = 131; Panel (**D**), N = 71).

**Table 1 genes-14-00525-t001:** Characteristics of subjects and VO2max, klotho levels.

Klotho (*n* = 202)	TRND		SED	
Number of patients	131		71	
	Men	Women	Men	Women
	80	51	27	44
Age (years), Mean ± SD	59.14 (±10.8)	57.24 (±9.4)	55.63 (±13.4)	61.91 (±10.5)
VO_2_ max (mL/kg/min)	42.27 (±9.2)	41.46 (±8.4)	38.25 (±7.9)	32.55 (±6.6)
Serum Klotho (pg/mL)	5.92 (±0.09)	5.39 (±0.15)	6.13 (±0,23)	5.71 (±0.15)

## Data Availability

Data are available upon request.

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
