# Peer review of "The Circulating Level of Klotho Is Not Dependent upon Physical Fitness and Age-Associated Methylation Increases at the Promoter Region of the Klotho Gene"

_genes, 2023, doi:10.3390/genes14020525_

Round 1

Reviewer 1 Report

Thanks to the respected authors, it was a thorough and comprehensive study.

The abstract of article has been completely summarized. Introduction is comprehensive and well-written. Method section is precise and comprehensive and discussion is well-founded.

The first paragraph of the discussion, I propose, should be devoted to the study's final findings.

The following articles are recommended for inclusion in the discussion section:

Prud'homme GJ, Kurt M, Wang Q. Pathobiology of the Klotho Antiaging Protein and Therapeutic Considerations. Front Aging. 2022 Jul 12;3:931331. doi: 10.3389/fragi.2022.931331. PMID: 35903083; PMCID: PMC9314780.

Natalia F, Ayu M, Afida KL, Arrianti EE, Meylyn L, Shine NW, Adiatmika IP. AEROBIC EXERCISE INCREASES COGNITIVE FUNCTION BIOMARKER, KLOTHO PROTEIN, IN ALZHEIMER DISEASE.

Author Response

Response to Reviewer 1

Dear Reviewer, thank you for comments and contribution.

Thanks to the respected authors, it was a thorough and comprehensive study.

The abstract of article has been completely summarized. Introduction is comprehensive and well-written. Method section is precise and comprehensive and discussion is well-founded.

The first paragraph of the discussion, I propose, should be devoted to the study's final findings.

Dear Reviewer, we made the changes, thank you for the recommendation.

The following articles are recommended for inclusion in the discussion section:

Prud'homme GJ, Kurt M, Wang Q. Pathobiology of the Klotho Antiaging Protein and Therapeutic Considerations. Front Aging. 2022 Jul 12;3:931331. doi: 10.3389/fragi.2022.931331. PMID: 35903083; PMCID: PMC9314780.

Natalia F, Ayu M, Afida KL, Arrianti EE, Meylyn L, Shine NW, Adiatmika IP. AEROBIC EXERCISE INCREASES COGNITIVE FUNCTION BIOMARKER, KLOTHO PROTEIN, IN ALZHEIMER DISEASE.

Dear Reviewer, thank you, we added them.

Reviewer 2 Report

The manuscript “The circulating level of klotho is not dependent upon physical fitness and age-associated methylation increases at the promoter region of the klotho gene” described the correlation between circulating klotho expression and physical fitness or KL promoter methylation. They concluded that circulating klotho was not related to fitness and methylation via a series of correlation analyses derived from blood samples. The research is novelty and meaningful, however, I think the data is not enough to support their conclusion. 

Some details showed be checked, such as the punctuation in lines 165-166.

Some paper has reported that promoter methylation is an important epigenetic mechanism of soluble Klotho reduction. In this research, the Pearson’s correlation can not exclude the role of methylation in soluble klotho expression well.

The author should describe more details in the methods. The Klotho promoter enriched of many CpGs, the standards to judge the methylation is not clear.

Why study the correlation between klotho and grip max?

 Klotho is aberant expressed in kidney diseases. Diseases should be considered for the sample selection.

Author Response

Response to Reviewer 2

Thank you for your constructive comments. We have revised the manuscript according to your suggestions.

The manuscript “The circulating level of klotho is not dependent upon physical fitness and age-associated methylation increases at the promoter region of the klotho gene” described the correlation between circulating klotho expression and physical fitness or KL promoter methylation. They concluded that circulating klotho was not related to fitness and methylation via a series of correlation analyses derived from blood samples. The research is novelty and meaningful, however, I think the data is not enough to support their conclusion.

Some details showed be checked, such as the punctuation in lines 165-166.

Dear Reviewer, thank you for your comments, we have corrected our paper according to your suggestions.

Some paper has reported that promoter methylation is an important epigenetic mechanism of soluble Klotho reduction. In this research, the Pearson’s correlation can not exclude the role of methylation in soluble klotho expression well.

Dear Reviewer, thank you and you are completely right and we do not exclude what you are suggesting.

The author should describe more details in the methods. The Klotho promoter enriched of many CpGs, the standards to judge the methylation is not clear.

Dear Reviewer, thank you we just followed the protocol of earlier studies.

Why study the correlation between klotho and grip max?

Dear Reviewer, handgrip strength is a strong indicator of total body muscle strength (Morley et al. 2001). Grip force is also predictive of disability and mortality (Rantanen et al. 1999, 2003). Semba et al also revealed an association between grip strength and plasma KL level in the elderly (Semba et al., 2011). In this study, we examined the correlation between klotho and grip max in adults.

Klotho is aberrant expressed in kidney diseases. Diseases should be considered for the sample selection.

Dear Reviewer, thank you for this comment. Kidney diseases were asked by the questionnaire and volunteer with kidney disease were disqualified.

Round 2

Reviewer 2 Report

The authors addressed some of my concerns. It can be accepted after carefully check.